# A Multi-Objective Permanent Basic Farmland Delineation Model Based on Hybrid Particle Swarm Optimization

**Hua Wang [1,\*], Wenwen Li [1], Wei Huang [1] and Ke Nie [2]**

1   School of Computer and Communication Engineering, Zhengzhou University of Light Industry, Zhengzhou 450002, China; 331707040335@zzuli.edu.cn (W.L.); hnhw235@zzuli.edu.cn (W.H.)
2   Key Laboratory of Urban Land Resources Monitoring and Simulation, Ministry of Natural Resources, Shenzhen 518034, China; nieke@whu.edu.cn
\*   Correspondence: 2013016@zzuli.edu.cn; Tel.: +86-18137810605

**Abstract:** The delimitation of permanent basic farmland is essentially a multi-objective optimization problem. The traditional demarcation methods cannot simultaneously take into account the requirements of cultivated land quality and the spatial layout of permanent basic farmland, and it cannot balance the relationship between agriculture and urban development. This paper proposed a multi-objective permanent basic farmland delimitation model based on an immune particle swarm optimization algorithm. The general rules for delineating the permanent basic farmland were defined in the model, and the delineation goals and constraints have been formally expressed. The model introduced the immune system concepts to complement the existing theory. This paper describes the coding and initialization methods for the algorithm, particle position and speed update mechanism, and fitness function design. We selected Xun County, Henan Province, as the research area and set up control experiments that aligned with the different targets and compared the performance of the three models of particle swarm optimization (PSO), artificial immune algorithm (AIA), and the improved AIA-PSO in solving multi-objective problems. The experiments proved the feasibility of the model. It avoided the adverse effects of subjective factors and promoted the scientific rationality of the results of permanent basic farmland delineation.

**Keywords:** permanent basic farmland; multi-objective; spatial optimization; particle swarm optimization; artificial immune algorithm; Xun County

## 1. Introduction

Over the past 60 years, the world has been undergoing rapid urbanization. Since the 1960s, the more rapid the economic development and the accelerated urbanization process in a region was, the more serious the decline in cultivated land was in the region [1–5]. In particular, some developing countries are facing the problems of population growth, increasing food demand, and limited agricultural production [6]. As a developing country with accelerating urbanization and industrialization in recent years, China has experienced rapid expansion in the urban construction area, and the loss in cultivated land area has also increased year by year. Today, China's basic national conditions are a large population, low per capita cultivated land resources, and insufficient resource reserves of cultivated land. To ensure food security and promote the development of modern agriculture, China has always attached great importance to the protection of farmland and began planning for basic farmland needs long ago. According to the requirements of the overall plan for national land area and the needs of the population, economy, and society, a certain percentage of the high-quality farmland on cultivated lands is zoned as a basic farmland protection area. For nearly 40 years, the issue of arable land protection

has always been of paramount importance. The new land management law further upgrades basic farmland to permanent basic farmland [7]. Compared with basic farmland, permanent basic farmland has a more prominent status, stricter protections, a more stable spatial pattern, and a more important position and role in stabilizing food production. The new law will ensure that by 2020, China's basic farmland protection will take shape in a relatively complete, powerful, effective, and orderly manner to protect food security in China [8].

Currently, to solve the problem of cultivated land loss, countries around the world have conducted related research. Mazzocchi et al. [9] used sensitivity index of agricultural land (SIAL) to understand the relationship between urban planning and agricultural land and analyzed the potential risks of shifting from cultivated land to construction land. Through case studies, the tool showed that external factors unrelated to agricultural activities were the main drivers of this shift. Terres et al. [10] studied and evaluated the main drivers of farmland abandonment in the 27 member states of the European Union. Through analysis, these drivers mainly included lower farm stability and viability, smaller farm size, and a negative regional context. In South Korea, the United States, India, and China, some scholars [11–16] have confirmed through research that rapid urbanization has accelerated the transformation of agricultural land to non-agricultural land, increasing the loss of arable land.

With regard to the delineation of permanent basic farmland, scholars from various countries have carried out many studies and simultaneously deepened their understanding of basic farmland. In general, the delimitation process should take into account the status of land use and the existing land constraints. In addition, attempts should be made to allocate permanent basic farmland protection to land with complete infrastructure and good natural resource conditions. Cultivated land that is spatially concentrated and not easily occupied by urban or industrial development can maximize the comprehensive benefits of arable land protection. Therefore, this can be regarded as a multi-objective spatial optimization problem. Based on the existing research, the methods of delimiting permanent basic farmland can be divided into three categories. In the first category, permanent basic farmland is demarcated mainly through the allocation of administrative indicators, which only need to meet the quota quantity requirements. Although some simple guidelines are given in the legislation, allocations are usually made in a more casual manner due to the lack of quantitative standards and a scientifically effective framework [17]. The second category introduces scientific evaluation systems and model analysis methods. Yang et al. [18] selected indicators from site conditions, transportation locations, agricultural production, and spatial morphology, constructed a comprehensive multifactor evaluation system, and delineated permanent basic farmland based on the principle of optimal comprehensive score. Liu et al. [19] used an analytic hierarchy process (AHP) network combined with local spatial autocorrelation analysis of basic farmland to determine the index weights. Zhang et al. [20] combined the spatial analysis functions of ArcGIS and the improved land evaluation and site assessment (LESA) method. Cheng et al. [21] used mathematical morphological image processing and GIS analysis technology to establish an evaluation index system and develop an analysis model for spatial patterns of farmland morphology. Most of the comprehensive considerations have been based on the natural conditions [20,22], economic benefits [23], and utilization levels [20,24] of cultivated land. The above methods have improved the rigor and scientificity of the delineation of permanent basic farmland to some extent. However, the selection of most indicators and the weighting of indicators still have a certain degree of subjectivity, and their dependence on the indicator system is too strong to solve a multi-objective optimization problem. The third method uses artificial intelligence algorithms that have emerged in recent years. Liu et al. [25] combined remote sensing, GIS, and artificial immune systems (AIS) and made some modifications to traditional artificial intelligence algorithms to divide farmland protection areas under spatial constraints. Zeng et al. [26] established a decision model system through the technique for order preference by similarity to an ideal solution (TOPSIS) method and constraint factors and classified farmland according to the proximity of the cultivated land to the ideal solution. Ma et al. [27] searched for protected areas through a seed expansion algorithm and

used human neural networks to predict the protection pressure of basic farmland. These attempts provided new directions for the delineation of permanent basic farmland.

In recent years, artificial intelligence algorithms have had great advantages in solving multi-objective spatial planning problems and have been successfully applied in problems such as land resource allocation [28–33], spatial pattern optimization [34–40], space selection [41–43], and land use zoning [44–47]. Among the artificial intelligence algorithms, particle swarm optimization (PSO) is a swarm intelligence algorithm that is inspired by bird foraging activities. As an evolutionary algorithm, PSO has the advantages of better ability to solve complex problems, higher convergence speed, and lower problem dimension. It is very suitable for high-dimensional spatial optimization problems, has a fast convergence speed, and can solve multi-objective optimization problems in multiple fields. However, the particle swarm algorithm has the disadvantages of premature convergence and the possibility of sinking into the local optima of a basic PSO algorithm when solving the spatial optimization problem [48]. The artificial immune algorithm (AIA) simulates the biological immune process and can ensure the diversity of individuals, help improve the global convergence ability of the particle swarm algorithm, and prevent particles from falling into the local optimal solution.

Therefore, an improved particle swarm optimization algorithm is proposed to solve the multi-objective optimization problem of permanent basic farmland delimitation. Xun county, a major grain production base in Henan province, was chosen as the study area for this study. Section 2 introduces the study area and data acquisition and processing. Section 3 describes the methodology of a multi-objective optimization model of permanent basic farmland delimitation based on the hybrid PSO algorithm. It defines the general rules and objective function of permanent basic farmland delineation, as well as the formal representation of the constraints, and discusses the improvement of the PSO algorithm when combined with the artificial immune algorithm. In Section 4, we describe the controlled trials that emphasized different subobjectives and compare the performances of the three models (PSO, AIA, and improved AIA-PSO) in solving the problem of permanent basic farmland delineation. Finally, Section 5 concludes the paper.

## 2. Data

### 2.1. Overview of The Study Area

To verify the feasibility of the model, we selected Xun County as the research area. The study area (35°26′00″ ~ 35°50′42″N, 114°14′52″ ~ 114°45′12″E) is located in northern Henan Province and eastern Hebi city and has an area of 954.98 km$^2$ (Figure 1). The geographical location of Xun County is shown in Figure 1. The county has 9 townships, 476 villages, and 78,711.18 hectares of agricultural land, of which 70,994.60 hectares are arable land. The county's overall topography is relatively flat—most of the areas are plains, and there are some hills and valleys in the west. Xun County has a provincial-level modern agricultural industrial park with an annual grain output of more than 1 million tons. It has been said since ancient times that "Li Yang harvest can take care of everyone".

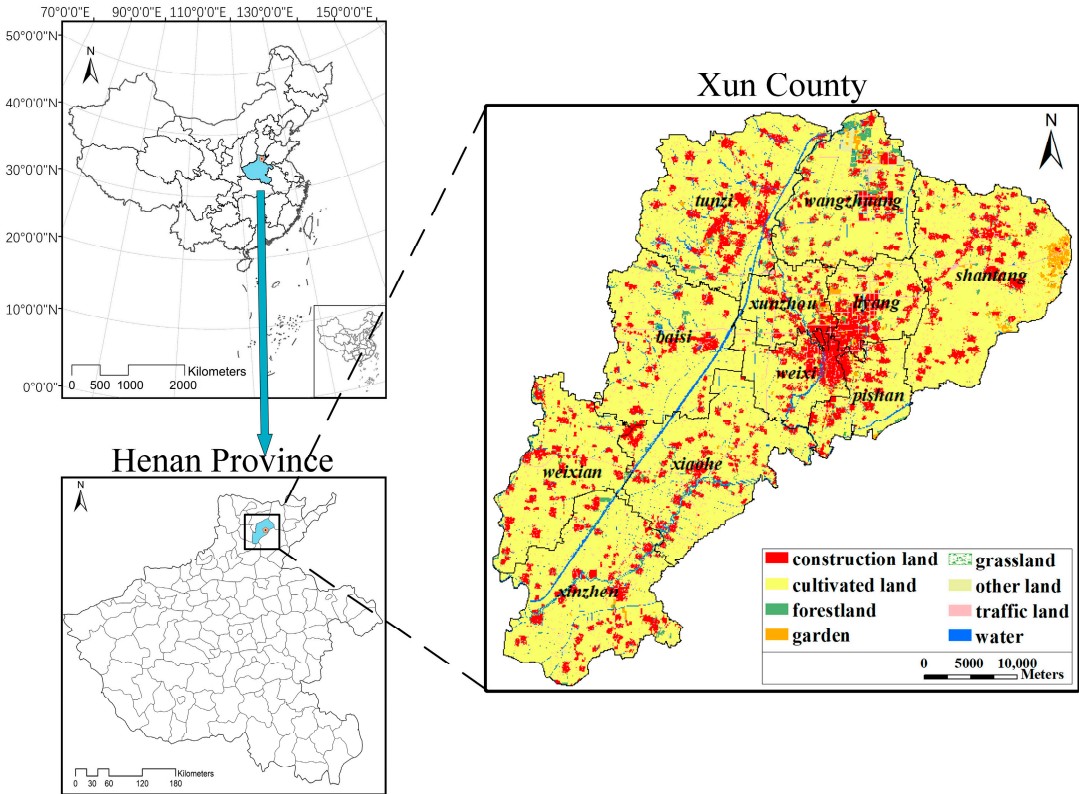

**Figure 1.** Geographical location and main utilization statuses of Xun County.

## 2.2. Data Processing

The purpose of delimiting the permanent basic farmland is to select arable land with high soil quality to facilitate farming and ensure that the area is protected and not easily occupied by urban expansion. The main influencing factors include infrastructure conditions, field conditions, soil fertility, and location conditions. This study needed data that included arable land quality, land use conditions, topography, climate, water conservancy and transportation conditions, etc., as shown in Table 1. To ensure the consistency of the data, the coordinate system of each layer in the study area was projected into CGCS2000_3_Degree_GK_Zone_38 using ArcGIS 10.2. The vector data were converted into raster data, and the corresponding values of the different influencing factors of the same raster cell were obtained. To ensure that the grid unit size was moderate, $100m \times 100$ *m* was used as the minimum grid unit. The total number of grid units in the study area was 98,360, of which the number of cultivated land units was 73,122. Then, the raster cells were encoded to add fields to store the corresponding data and allow for numerical mapping. The spatial data of the main impact factors are shown in Figure 2.

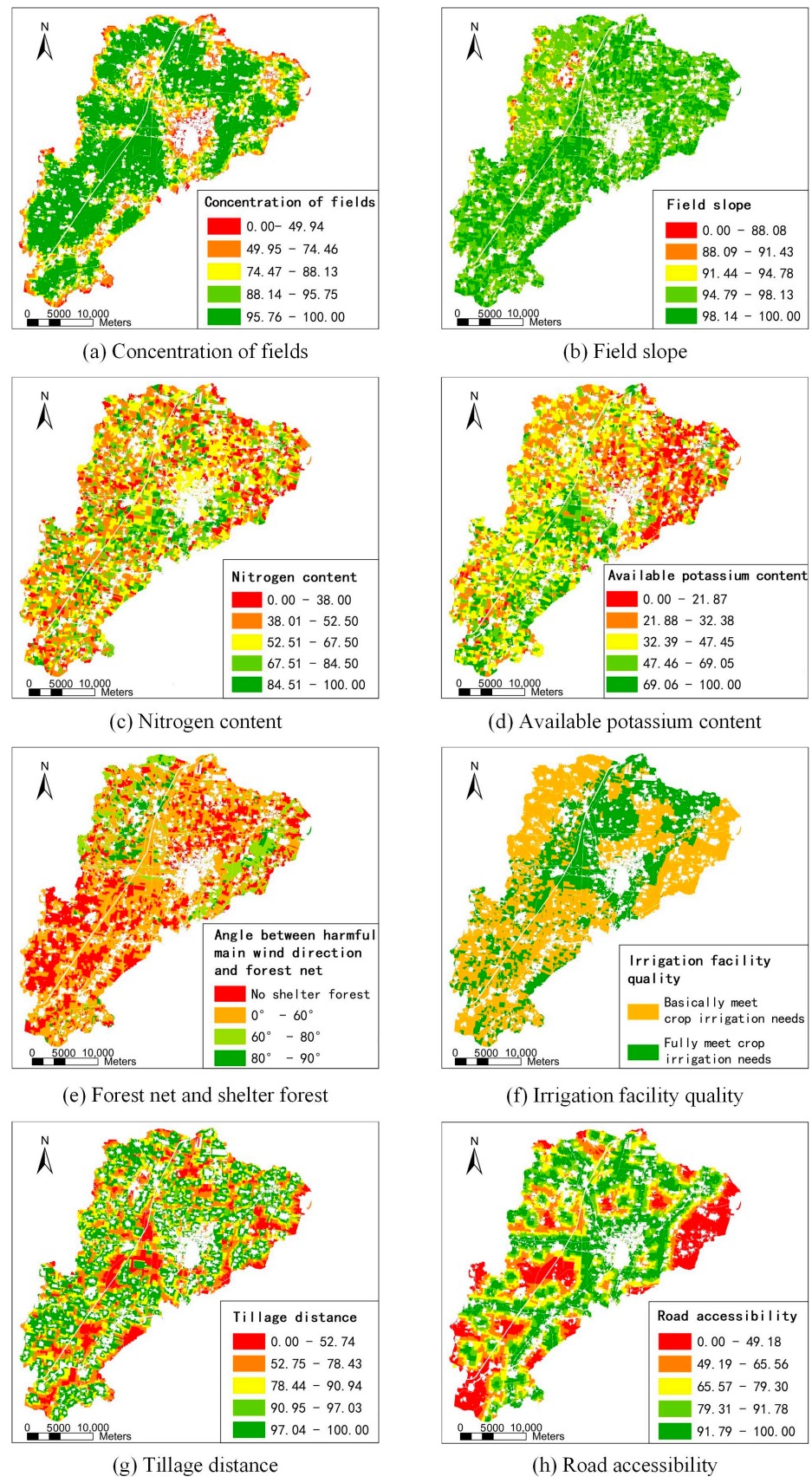

**Figure 2.** Spatial data of the main influencing factors in Xun County: (**a**) concentration of fields, (**b**) field slope, (**c**) nitrogen content, (**d**) available potassium content, (**e**) net forest and shelter forest, (**f**) irrigation facility quality, (**g**) tillage distance, and (**h**) road accessibility.

**Table 1.** Description of data sources.

| No. | Data | Description of Data Source | Department of Data Source |
|---|---|---|---|
| 1 | Soil fertility, topography and landforms, irrigation facilities, field shape, field size | The latest results of the cultivated land quality grades, cultivated land quality update evaluation results, and agricultural land classification results | Agriculture Bureau |
| 2 | Land use type | The land use change survey results, the results of the third national land survey, three-line delineation results and data | Natural Resources Bureau |
| 3 | The forest net and shelter forest | The forestry resource survey results | Forestry Bureau |
| 4 | Annual average temperature, annual average precipitation | Meteorological monitoring results | Meteorological Bureau |
| 5 | Road accessibility | Road distribution maps, traffic planning, and layout information | Traffic and Transportation Bureau |
| 6 | Ecological red line constraint | The "13th Five-Year Plan" for environmental protection | Environmental Protection Bureau |
| 7 | Town bound boundaries | Industrial and tourism development planning | Development and Reform Commission |

## 3. Methodology

### 3.1. Multi-Objective Optimization Model of Permanent Basic Farmland Delimitation

#### 3.1.1. Objective Functions

Permanent basic farmland delineation was based on the local natural endowment and was combined with the area constraints, farming environment, and site conditions to optimize the selection of cultivated land units. To achieve the final delineation of the permanent basic farmland protection area, which can achieve the objectives of having better arable land quality and complete infrastructure, being centralized and continuous, and not being easily occupied, this study set the target system for the permanent basic farmland delineation model as the following three subobjectives: the land suitability objective, land continuity objective, and land stability objective.

Land suitability objective:

The quality of cultivated land is a combination of factors such as natural conditions, economic factors, and water conservancy facilities of cultivated land, and is one of the important reference standards for the delineation of permanent basic farmland.

This article refers to the requirements in the "Regulation for Gradation on Agriculture Land Quality (GB/T 28407-2012)" ("Regulation for Gradation on Agriculture Land Quality (GB/T 28407-2012)" was released by China Land Consolidation and Rehabilitation in 2014 (http://lcrc.org.cn/tdzzgz/bzhjs/gjbz/201412/t20141204_27731.html).) and "Cultivated Land Quality in Henan Province in the Annual Update Evaluation Work Training Materials", which is provided by the Department of Natural Resources of Henan Province, and according to the "Cultivated Land Quality Analysis Report of Xun County in 2018", which is provided by the Natural Resources Bureau of Xun County. Henan Province's cultivated land use index was divided into 19 grades, with Grade 1 being the highest grade. The corresponding land use of Xun County was 5, 6, 7, and 8. For the convenience of calculation, the cultivated land use equalization was mapped to the range of [0,1] to obtain $suit_{ij}$, as shown in Table 2. Based on this value and the decision variable $r_{ij}$ of the selected grid cell, the overall suitability could be calculated. Since this model used grid spatial data, there were two possibilities for the decision state of each land grid unit. $r_{ij} = 1$ indicated that the grid cell was selected. If the cell was not selected, $r_{ij} = 0$. Therefore, it was only necessary to calculate the unit in which the decision variable was 1. The use of $f_s$ to represent the objective of land suitability can be described as Formula (1).

$$f_s = \sum_{i=0}^{m} \sum_{j=0}^{n} r_{ij} \cdot suit_{ij} \tag{1}$$

**Table 2.** Agricultural land quality rating mapping table.

| Land Use Grade | 5 | 6 | 7 | 8 |
|---|---|---|---|---|
| Mapped value | 1.0 | 0.7 | 0.4 | 0.1 |

In the formula, $m$ is the total number of grid rows, $n$ is the total number of grid columns, and $i$ and $j$ are the values of the row and column. The purpose of the land suitability objective is to maximize the value of $f_s$ as much as possible. The cultivated land units with better conditions in various aspects in the study area were classified into basic farmland protection areas.

Land continuity objective:

Spatial continuity refers to the proximity of the spatial locations of the different farmland units selected for basic farmland within the same area. Scattered basic farmland affects the farmland's efficiency in cultivation, fertilization, and transportation, increases labor costs, and is detrimental to food production. In addition, contiguous lands require less infrastructure and other services to facilitate management.

To avoid fragmentation of the overall pattern of basic farmland, this study calculated the degree of continuous land use of the basic farmland with Formula (2). In the formula, $L_{sum}$ is the sum of the perimeter of the delineated permanent basic farmland. When the defined area is circular, the permanent basic farmland is the most concentrated, and the minimum perimeter $L_{min}$ can be calculated. Conversely, if the selected units are far apart, the perimeter is the largest, and this value is $L_{max}$. $f_l$ can indicate the degree of concentration of land in the study area. The larger the difference between $L_{max}$ and $L_{sum}$ is, the smaller the perimeter sum of the permanent basic farmland, and the larger the $f_l$ is, the higher the continuity of the selected basic farmland.

$$f_l = \frac{L_{max} - L_{sum}}{L_{max} - L_{min}} \tag{2}$$

Land stability objective:

Land stability can be understood as the possibility that land will not be occupied or misappropriated within a certain period of time. In recent years, a large number of people and resources have flowed into cities and towns, which has led to the rapid expansion of cities and towns into the surrounding areas, that has a negative impact on the stability of surrounding farmland. The cultivated land near the town center and the main transportation roads are subject to more occupation pressure due to low development costs and convenient transportation. However, the protection of cultivated land should not restrict the development of the local economy. In the delineation process, this was used as a negative factor to improve the stability of permanent basic farmland.

As in Formula (3), the land development potential of each unit in the grid depended on the Euclidean distance between the grid unit and the nearest road, railway, highway, and town center, which are expressed in the formula as $DRoads_{ij}$, $DRailways_{ij}$, $DExpressway_{ij}$, and $DDistrict_{ij}$, respectively. The corresponding weights $\alpha_1$, $\alpha_2$, $\alpha_3$, and $\alpha_4$ respectively, satisfy $\alpha_1 + \alpha_2 + \alpha_3 + \alpha_4 = 1$ and normalize $d_{ij}$ to $D_{ij}$, which is convenient for subsequent calculations, such as in Formula (4). The study assigned 0.3, 0.1, 0.1, and 0.5 to $\alpha_1$, $\alpha_2$, $\alpha_3$, and $\alpha_4$ respectively, according to the road levels, impacts, and occupation possibilities [49].

$$d_{ij} = \alpha_1 \cdot DRoads_{ij} + \alpha_2 \cdot DRailways_{ij} + \alpha_3 \cdot DExpressway_{ij} + \alpha_4 \cdot DDistrict_{ij} \tag{3}$$

$$D_{ij} = \frac{d_{ij} - d_{min}}{d_{max} - d_{min}} \tag{4}$$

where the maximum and minimum values that $d_{ij}$ can reach are $d_{max}$ and $d_{min}$, respectively. The larger the values of $DRoads_{ij}$, $DRailways_{ij}$, $DExpressway_{ij}$, and $DTown_{ij}$ are in Formula (4), the farther the unit is from nearby towns and transportation centers, and the larger the corresponding value of $D_{ij}$,

which indicates that the land unit is more stable and more suitable for being classified as permanent basic farmland. The sum of the product of $D_{ij}$ and the decision variable $r_{ij}$ can represent the land stability $f_w$, as shown in Formula (5):

$$f_w = \sum_{i=0}^{m} \sum_{j=0}^{n} r_{ij} \cdot D_{ij} \tag{5}$$

In summary, it is clear that the delineation of the permanent basic farmland is a multi-objective optimization problem. For the above three subobjectives, the model chooses a linear weighting method to process the objective function $f(x)$, as shown in Formula (6). Max{} refers to the maximum value of the linear weighting function in parentheses.

$$f(x) = Max\{\rho_1 \cdot f_s + \rho_2 \cdot f_l + \rho_3 \cdot f_w\}$$
$$\rho_1 + \rho_2 + \rho_3 = 1 \tag{6}$$

### 3.1.2. Constraint Conditions

1. Constraints on the total area of selected cultivated land

The target of the permanent basic farmland protection provided to Xun County by the general utilization plan for land and space in Hebei city was 60,246.67 hectares, with a protection rate of 84.86%. $Area_{Selected}$ indicates the total farmland area selected in Xun County, which must satisfy Formula (7), where $M_{ij}$ is the area of each land unit.

$$Area_{Selected} = \sum_{i=0}^{m} \sum_{j=0}^{n} r_{ij} \cdot M_{ij} \geq 60246.67 \tag{7}$$

2. Land use constraints

Due to the historical rules of land use conversion, it is difficult to convert urban construction land into agricultural land, and it cannot be used as an alternative permanent basic farmland. For this model, the spatial units of urban construction land were eliminated in advance and were not involved in the model calculation.

3. Constraints on topographical conditions

According to the "notice on strengthening and improving the protection of permanent basic farmland" of the Ministry of Natural Resources, when the slope of the cultivated land is $\geq 25°$, the slope is too steep, which greatly limits the use of such cultivated land. It is necessary to gradually return steeply sloped farmland to forest and grass. For this type of land, the model marks it as $code_{ij} = 0.9$, with the average slope value of the unit represented by $Slope_{ij}$, as shown in Formula (8):

$$Code_{ij} = Slope_{ij} > 25?0.9 : Code_{ij} \tag{8}$$

4. Urban boundary constraints

Prioritizing eligible cultivated land at the borders of cities and towns into permanent basic farmland can help cities rationalize their spatial layout, promote urbanization in an orderly manner, and save land resources while also allowing full utilization of the basic farmland's multiple functions of protecting natural ecology, optimizing spatial structure, reducing urban pollution, and beautifying the urban environment. We used $DRoads_{ij}$ to represent the distance from the land unit to the nearest road and $DTown_{ij}$ to represent the distance between the unit and the nearest town center. In the calculation of $DRoads_{ij}$ and $DTown_{ij}$ for each land unit and weighting of the sum, as in Formula (9), the smaller

the value was, the greater the probability that the corresponding land unit would be classified as permanent basic farmland. According to the possibilities and impacts of construction land expansion, we assigned 0.3 and 0.7 to $\sigma_1$ and $\sigma_2$, respectively [49].

$$T_{ij} = \sigma_1 \cdot DRoad_{ij} + \sigma_2 \cdot DTown_{ij}$$
$$\sigma_1 + \sigma_2 = 1$$

$$(9)$$

### 3.2. Design of AIA-PSO Model

#### 3.2.1. Particle Encoding and Initialization

Each particle in the PSO is considered a candidate solution for the permanent basic farmland delineation problem. The particle guides its own position and speed continuously according to the best position of the individual particle it passes through and the global best position experienced by the entire group. To facilitate the determination and updated location of the particle's subsequent decision variables, binary coding and real number coding were used, respectively.

There were only two possible values for each candidate unit in the raster data. Binary coding was used to mark whether the grid cell was selected as permanent basic farmland. As shown in Figure 3. If the cell was not involved in the calculation, it was marked as "null" (represented by N in the Figure 3). For the positions and velocities of the particles, the encoding method of the particles is shown in Figure 3b. In addition, this model defines a particle structure, which contains not only information about the delineation scheme but also information about the particle fitness, the particle's current position, the individual best position, and the global best position, as shown in Figure 3c.

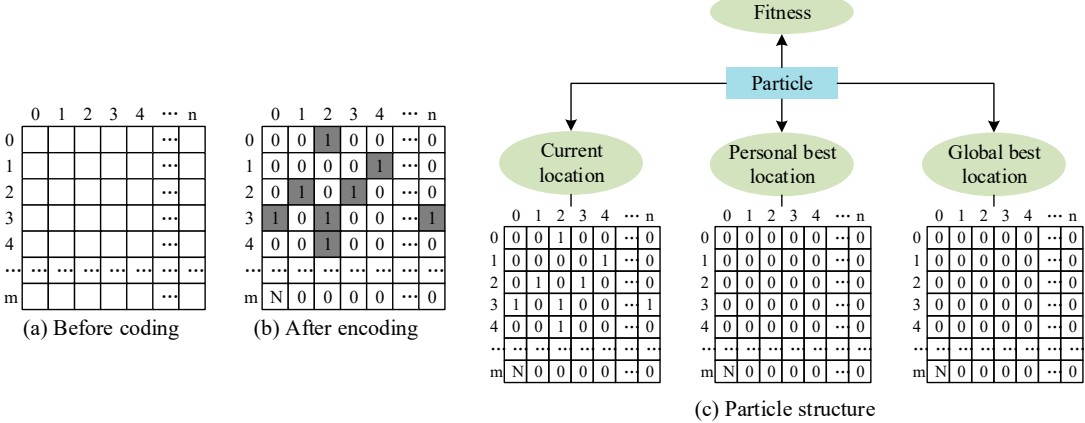

**Figure 3.** Particle coding strategy: (**a**) grid cells state before particle encoding; (**b**) grid cells state after particle encoding; (**c**) Particle structure.

When the particles were initialized, if particles were generated completely randomly, some restricted land units could be selected into permanent basic farmland. Therefore, during the initialization process, it was necessary to consider the limitation of the introduced constraint conditions and use a random mode with a constraint to initialize the particles.

In Section 3.2, when the unit slope was greater than $25°$, we used $Code_{ij} = 0.9$ to mark the restricted land unit. When the particle starts to initialize, the value of the decision variable $r_{ij}$ is determined according to $Code_{ij}$. First, the randint() random function was used to generate $u_s$, which is a random number between 0 and 1. If $u_s = 1$, a random $u_s$ was generated again in the same way. If $u_d > Code_{ij}$, $r_{ij} = 1$, otherwise, $r_{ij} = 0$. If $u_s = 0$, then $r_{ij} = 0$. With this method of generating random numbers, the probability of $u_d > Code_{ij}$ was very small, and the probability that the unit was selected into permanent basic farmland will be reduced accordingly.

### 3.2.2. Particle Position and Velocity Update

During the search process, particles continuously update their speed and position according to the individual optimal solution and the global optimal solution. $v(t)$ represents the speed of the particle at the t-th iteration, and $x(t)$ represents the position of the particle at the t-th iteration. The updated formula is expressed in Formula (10). In the formula, $\omega(t)$ is the dynamic inertia weight, $\beta_1$ and $\beta_2$ are the acceleration constants, that is, the individual learning factors and social learning factors of the particles, $a_1$ and $a_2$ are the independent random numbers between [0,1], and $p_{ij}(t)$ and $p_{gj}(t)$ are the individual optimal solutions and global optimal solution of the particles.

$$
\begin{cases}
v_{ij}(t+1) = & \omega(t)v_{ij}(t) + \beta_1 a_1\left(p_{ij}(t) - x_{ij}(t)\right) + \beta_2 a_2\left(p_{gj}(t) - x_{ij}(t)\right) \\
x_{ij}(t+1) = x_{ij}(t) + v_{ij}(t+1) \\
\omega(t) = \omega_{max} - \frac{t}{iternum} \cdot (\omega_{max} - \omega_{min})
\end{cases}
\tag{10}
$$

where $\omega_{max}$ is the set maximum weight value of $\omega(t)$, $\omega_{min}$ is the set minimum weight value, *iternum* is the maximum number of iterations, $t$ is the current number of iterations, and they jointly determine $\omega$ (t). In general, $\omega(t)$ changes linearly and gradually. In this way, the global search ability of the algorithm was strengthened in the early stages of the iteration, and the local search ability was more focused in the later stages. As the number of iterations increases, $\omega(t)$ gradually decreases, and the impact of the speed of the previous iteration on the current speed is smaller.

Due to the different coding methods of the model for the particle position and decision variables, the updated particle cannot be guaranteed to be 0 or 1, so it was necessary to improve the position update mechanism. Therefore, the fuzzy function $Sigmoid(x)$ was introduced. Since the value of $x_{ij}$ is 0 or 1 is determined by the speed, the fuzzy function $Sigmoid(x)$ as expressed as shown in Formula (11), and the position update formula was changed to Formula (12):

$$
Sigmoid\left(v_{ij}\right) = \frac{1}{1 + exp\left(v_{ij}\right)}
\tag{11}
$$

$$
x_{ij}(t+1) = \begin{cases}
0, & \varepsilon \geq Sigmoid\left(v_{ij}(t+1)\right) \\
1, & otherwise
\end{cases}
\tag{12}
$$

where $\varepsilon$ is a random number between 0 and 1. According to the above formula, the value range of $x_{ij}(t+1)$ is controlled to $\{0,1\}$. In the binary algorithm, $v_{ij}$ can be regarded as the probability. If the value of the probability $Sigmoid\left(v_{ij}\right)$ is 1, the value of the probability of $1 - Sigmoid\left(v_{ij}\right)$ is 0. The probability change of the particles can be expressed as Formula (13):

$$
\rho = Sigmoid\left(v_{ij}\right)\left(1 - Sigmoid\left(v_{ij}\right)\right)
\tag{13}
$$

For any particle in the model, $v_{max}$ was set as the maximum speed to prevent particles from flying out of the search area, which was also in the allowable probability range of the binary representation algorithm. By calculation, when $v_{max} > 10$, $Sigmoid(v_{max}) < 4.53 \times 10^{-5}$, then the position update of the algorithm was meaningless. Through the calculation of $Sigmoid\left(v_{ij}\right)$, to ensure that the position of the particles can be changed, $v_{max}$ is set to 6, at this time, $0.0025 < Sigmoid\left(v_{ij}\right) < 0.9975$.

### 3.2.3. Improvement of PSO with the Artificial Immune Algorithm

Particles are randomly assigned initial positions and initial velocities during the initialization process and continuously update their individual optimal positions and global optimal positions during the search process. This leads to the particles tending to be identical, which reduces the diversity, and they fall into a local optimal solution. The artificial immune algorithm is an intelligent search algorithm inspired by the biological immune system [50]. Its biggest feature is that it has a strong

global search ability and can maintain the diversity of antibodies, introducing immune system concepts into particle swarm optimization and using its diverse generation and maintenance mechanism of the immune system to overcome the premature convergence problem of particle swarm optimization. In the course of immunization, the selection of effective antibodies depends on the affinity between the antibody and the antigen. Through the "survival of the fittest" mechanism, antibodies with high affinity in the cloned population participate in reproduction and mutation, while the antibodies with low affinity are inhibited and gradually discarded with each iteration. Based on this immune principle, probability$_i$ was used to represent the selection probability of the i-th particle, and the formula is as follows:

$$probability_i = \frac{\sum_{j=1}^{N+N_0}|f(x_i)-f(x_j)|}{\sum_{i=1}^{N+N_0}\sum_{j=1}^{N+N_0}|f(x_i)-f(x_j)|} \quad i,j = 1,2,\cdots,N+N_0 \tag{14}$$

where N is the number of particles generated by the particle swarm algorithm, and $N_0$ is the number of new randomly generated particles.

To ensure the diversity of particles and the generation of new antibodies, the model performed immune cross-operation on the two antibodies and combines the best retention strategies to obtain better antibodies. As shown in Figure 4, two parent antibodies are selected in the particle population: particle A (P (A) in Figure 4) and particle B (P (B) in Figure 4). Since it is a replacement operation of a certain area between two particles, the same area is randomly selected in the two particles, and cross-replacement is performed to obtain new particles, P′(A) and P′(B). The fitness values of the parent antibody and the child antibody are calculated separately. If the fitness value of the hybridized child antibody is greater than the corresponding parent antibody, the corresponding parent antibody is replaced with the child antibody. Otherwise, it is not updated.

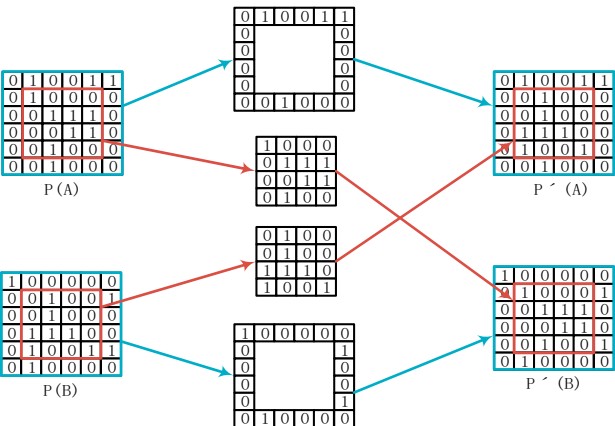

**Figure 4.** Particle immune crossover operation.

The particle updating in the particle swarm algorithm depends on the individual optimal solution to a large extent and the global optimal solution experienced by the particle, but the accuracy of this one-way search is not ideal. To cause the particles to more comprehensively search for optimal solutions, we chose to improve the particle speed updating formula according to the individual optimal solution of the first $N_b$ particles with the best fitness value in the population, such as Formula (15), which effectively improved the accuracy of the algorithm search.

$$v_{ij}(t+1) = \omega(t)v_{ij}(t) + \sum_{i=1}^{N_b}\beta_1 a_1\left(p_{ij}(t) - x_{ij}(t)\right) + \beta_2 a_2\left(p_{gj}(t) - x_{ij}(t)\right) \tag{15}$$

### 3.2.4. Fitness Function Design

PSO uses the fitness function to judge whether a particle is good or bad. A higher fitness value indicates that the particle is closer to the optimal solution. Under the various constraints, to solve the multi-objective optimization problem of permanent basic farmland delimitation, not only do the three functions for the subobjectives of land suitability, land continuity, and land stability need to be combined, but also the constraints, such as the size of the protected area and the boundaries of cities and towns, must be considered. Therefore, this model uses the most commonly used external penalty functions to deal with the constraints of permanent basic farmland delineation, which is transformed into a part of the objective function, which becomes an unconstrained optimization problem. The external penalty function is expressed in Formula (16):

$$\varphi(x) = f(x) \pm \left[ \sum_{i=1}^{p} \gamma_i G_i(x) + \sum_{j=p+1}^{q} c_j H_j(x) \right] \tag{16}$$

In Formula (16), $\gamma_i$ and $c_j$ are positive penalty coefficients, and $G_i(x)$ and $H_j(x)$ are functions of the inequality constraint $g_i(x)$ and the equality constraint $h_j(x)$, respectively. Generally, $\delta$ and $\gamma$ in the formula take values of 1 or 2.

$$G_i(x) = max[0, g_i(x)]^{\delta}$$
$$H_j(x) = \left| h_j(x) \right|^{\gamma} \tag{17}$$

### 3.2.5. Flow of AIA-PSO Model

Based on the above ideas and the construction of the permanent basic farmland objective function and the constraint condition system combined with the introduction of artificial immune concepts, and the particle swarm algorithm to improve the particle speed and location update strategy when solving the permanent basic farmland demarcation problem, the overall flowchart is shown in Figure 5.

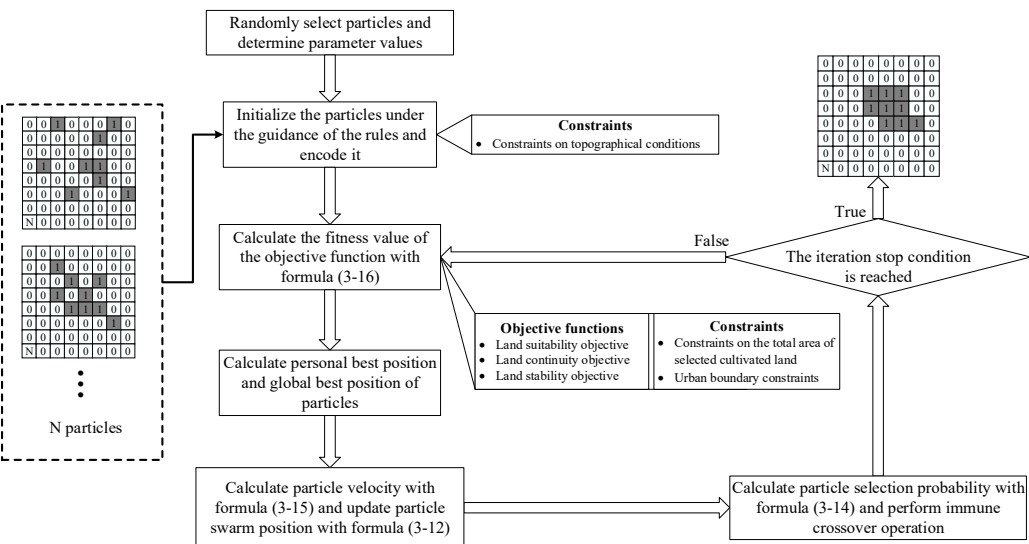

**Figure 5.** Model flow chart.

## 4. Analysis of the Permanent Basic Farmland Demarcation Results

### 4.1. Setting PSO Model Parameters

The parameters of the permanent basic farmland delineation model based on the particle swarm algorithm mainly included particle size $N$, maximum iterations *iternum*, inertial weight $\omega$, acceleration constants $c_1$ and $c_2$, random numbers $r_1$ and $r_1$, and maximum particle velocity $v_{max}$. Among these,

the inertia weight $\omega$ depended on its maximum value $\omega_{max}$, minimum value $\omega_{min}$, and maximum number of iterations, *iternum*. We referred to the typical parameter values of PSO proposed by Carlisle and Dozier [51] to set the main parameters of this model. The specific settings are shown in Table 3.

**Table 3.** Parameter value settings of the model.

| N | *iternum* | $\omega_{max}$ | $\omega_{min}$ | $\beta_1$ | $\beta_2$ | $a_1$ | $a_2$ | $v_{max}$ |
|---|---|---|---|---|---|---|---|---|
| 30 | 100 | 0.9 | 0.4 | 2.8 | 1.3 | [0,1] | [0,1] | 6 |

To eliminate land that was not suitable for basic farmland in advance and improve the calculation efficiency of the model, we reclassified the grid cells into six categories: cultivated land, other agricultural land, unused land, urban and rural construction land, transportation land, and water areas. During the model initialization process, the construction land and land units that did not meet the basic site conditions were preprocessed. These land units were not considered during the model operation.

*4.2. Comparison of the Different Schemes*

This model needs to be comprehensively considered from three aspects of land suitability, continuity, and stability. The weight coefficients corresponding to the subobjective functions were set in advance. To explore the difference in the impact of different weighting schemes on the delineation results of the permanent basic farmland protection areas, this paper set up three sets of optimization schemes based on three subobjective functions: quality assurance schemes (weight ratio of 8:1:1), spatial optimization schemes (weight ratio of 1:8:1), layout stability schemes (weight ratio of 1:1:8), as well as multiple sets of control experiments.

1. Comparative analysis with quality assurance schemes

Because the quality assurance comparison plan focused on optimizing the quality of cultivated land, scheme A sets the target weight of land suitability to 0.8, and the other two target functions had a weight of 0.1. Since permanent basic farmland should be high-quality agricultural land, and therefore the suitability of the weight cannot be 0, scheme D was set to only consider the suitability of land in extreme cases, and its weight was set to 1.00. In schemes E and F, the land continuity target and the land stability target were given higher weight values. In comparison with scheme A, the ratio of schemes E and F was set to 2:1:1. Scheme H balanced the three goals and assigned weight factors with a ratio of approximately 1:1:1. The above weight distribution schemes are shown in Table 4, and the experimental results of the objective function values of each subfunction are shown in Table 5.

**Table 4.** Combination scheme of weighting factors for quality assurance type.

| Scheme | Weight of Land Suitability $\rho_1$ | Weight of Land Continuity $\rho_2$ | Weight of Land Stability $\rho_3$ |
|---|---|---|---|
| D | 1.00 | 0.00 | 0.00 |
| A | 0.80 | 0.10 | 0.10 |
| E | 0.50 | 0.25 | 0.25 |
| H | 0.34 | 0.33 | 0.33 |

**Table 5.** Objective function values under different weighting schemes for quality assurance types.

| Scheme | Land Suitability $f_s$ | Land Continuity $f_l$ | Land Stability $f_w$ |
|---|---|---|---|
| D (1.0:0.0:0.0) | 39,353.9 | 0.7923 | 41,763.87 |
| A (0.8:0.1:0.1) | 39,027.4 | 0.8481 | 41,879.81 |
| E (0.5:0.25:0.25) | 36,891.8 | 0.8864 | 43,312.71 |
| H (0.34:0.33:0.33) | 34,988.9 | 0.9102 | 44,836.75 |

It can be seen from scheme D that when only the land suitability target was considered, $f_s$ reached a maximum value of 39,353.9, but the land continuity and land stability index were low. In a comparison of scheme A with schemes D, E, and H, as the continuity and stability weights increased, their corresponding function values increased by up to 14.88% and 7.06%, respectively. At the same time, the suitability values decreased by up to 11.09%, and the average level decreased from 6.17 to 6.52. The land suitability goals drive particles to choose land units with higher grades, but these land units were not necessarily continuous. However, some cultivated lands with higher grades were closer to urban roads, so as the weight ratio changed, the suitability index gradually decreased, and continuity and stability improved accordingly.

2. Comparative analysis of the spatial optimization schemes

In the spatial optimization comparison schemes, the focus was on the degree of continuity of the cultivated land, so scheme B set the weight of land continuity to 0.8, and the other two objective function weights to 0.1. In scheme F, based on the emphasis on continuity, the weights of land suitability and land stability were increased to analyze the impact of the change in weight on the continuity index. The above weighting scheme is shown in Table 6, and the running result of each subobjective function is shown in Table 7.

**Table 6.** Combination of weighting factors for spatial optimization scheme.

| Scheme | Weight of Land Suitability $\rho_1$ | Weight of Land Continuity $\rho_2$ | Weight of Land Stability $\rho_3$ |
|--------|------------------------------------|-----------------------------------|----------------------------------|
| B | 0.10 | 0.80 | 0.10 |
| F | 0.25 | 0.50 | 0.25 |
| H | 0.34 | 0.33 | 0.33 |
| D | 1.00 | 0.00 | 0.00 |

**Table 7.** Objective function values under different weighting schemes for spatial optimization scheme.

| Scheme | Land Suitability $f_s$ | Land Continuity $f_l$ | Land Stability $f_w$ |
|--------|------------------------|-----------------------|----------------------|
| B(0.1:0.8:0.1) | 33,828.2 | 0.9646 | 42,174.14 |
| F(0.25:0.5:0.25) | 34,314.2 | 0.9293 | 43,617.10 |
| H(0.34:0.33:0.33) | 34,988.9 | 0.9102 | 44,836.75 |
| D(1.0:0.0:0.0) | 39,353.9 | 0.7923 | 41,763.87 |

Scheme B was the spatial optimization scheme set in this study, and the land continuity index reached 0.9646. In a comparison of scheme B with schemes F and H, as the suitability and stability weights increased, the corresponding objective function values increased by up to 3.43% and 6.31% respectively, and the continuity of the land decreased by up to 5.64%. The continuous objective guides particles towards the area with concentrated space, but it inevitably falls into land with poor cultivated land quality or adjacent urban area activity, which leads to a decrease in the land suitability objective and cultivated land stability. A typical area in the study area was selected for observation and analysis, as shown in Figure 6. As the weight of the continuous target gradually increased, this area also produced a more obvious concentrated continuous effect.

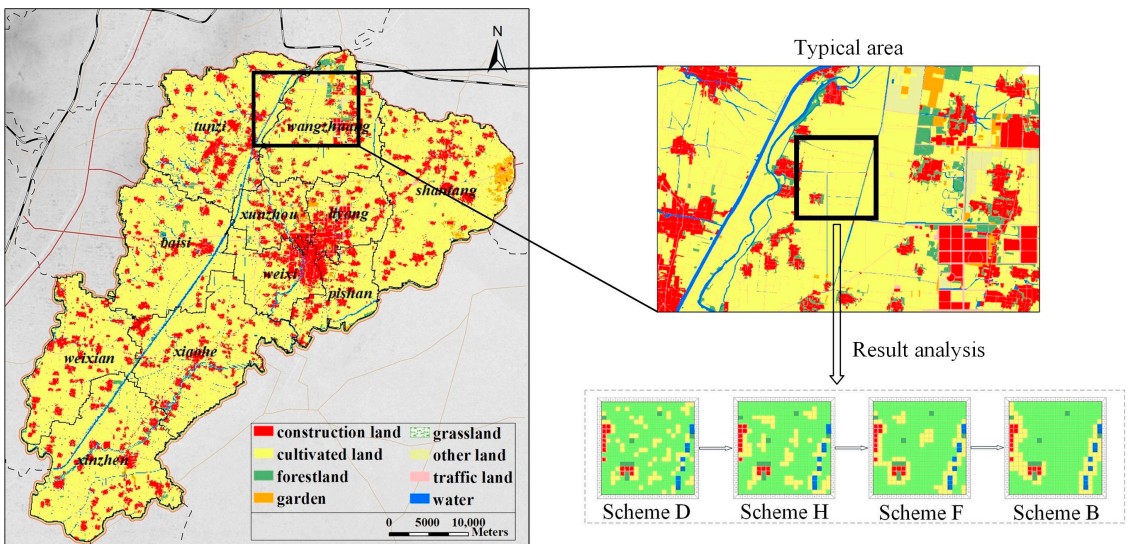

**Figure 6.** Changes in the spatial pattern of typical regions affected by contiguous weights.

## 3. Comparative analysis of the layout stability schemes

The layout stability comparison scheme mainly analyzed whether the cultivated land in the permanent basic farmland protection area could remain stable for a long time without being occupied. Therefore, in scheme C, the land stability parameter $\rho_3$ was set to 0.8, and the remaining two objective function weights were also set to 0.1. Schemes G and H were compared with scheme C to analyze the impacts of changes in the land suitability and continuity weights on the stability. On the basis of emphasizing the stability coefficient, the weights of the first two were increased to analyze their impact. The weighting schemes are shown in Table 8, and the running results of the objective values of each subobjective function are shown in Table 9.

**Table 8.** Combination scheme of weighting factors for the layout stability scheme.

| Scheme | Weight of Land Suitability $\rho_1$ | Weight of Land Continuity $\rho_2$ | Weight of Land Stability $\rho_3$ |
|---|---|---|---|
| C | 0.10 | 0.10 | 0.80 |
| G | 0.25 | 0.25 | 0.50 |
| H | 0.34 | 0.33 | 0.33 |
| D | 1.00 | 0.00 | 0.00 |

**Table 9.** Objective function values under the different weighting schemes for the layout stability scheme.

| Scheme | Land Suitability $f_s$ | Land Continuity $f_l$ | Land Stability $f_w$ |
|---|---|---|---|
| C (0.1:0.1:0.8) | 33,414.2 | 0.8273 | 49,230.57 |
| G (0.25:0.25:0.5) | 34,014.2 | 0.8775 | 47,161.21 |
| H (0.34:0.33:0.33) | 34,988.9 | 0.9102 | 44,836.75 |
| D (1.0:0.0:0.0) | 39,353.9 | 0.7923 | 41,763.87 |

Scheme C was the best layout stability scheme set in this article, and the stability index reached 49,230.57 for the study period. Comparing scheme C with schemes G and H, as the weights of suitability and continuity increase, their corresponding values increase by 4.71% and 10.02% respectively, and the stability decreased by up to 9%. The stability objective guides particles away from the areas with developed transportation and densely populated cities and towns, thereby avoiding the possibility of occupation of cultivated land, but these areas may conflict with the other two objectives. Increasing

the stability weight moves particles far away from towns and roads, but this contradicts the town boundary constraints, and the two restrict each other.

Two groups of typical areas were selected in the study area: the periphery of the town and the periphery of the main traffic road. Among these, the areas marked in pink are urban areas, and the areas marked in dark gray are road areas (shown in Figure 7). From the results of scheme D (with a stability weight of 0), scheme H (with a stability weight of 0.33), scheme G (with a stability weight of 0.5), and scheme C (with a stability weight of 0.8), it can be seen that when the stability weight values are small, the movement of the particle is more dependent on the quality of the land. At the same time, due to the limitation of urban boundaries, the selected farmland mostly gathered around cities and roads. As the weight of stability gradually increases, the selected area gradually moves away from the towns and roads, which has the effect of moving away from urban traffic and reducing the possibility of encroachment.

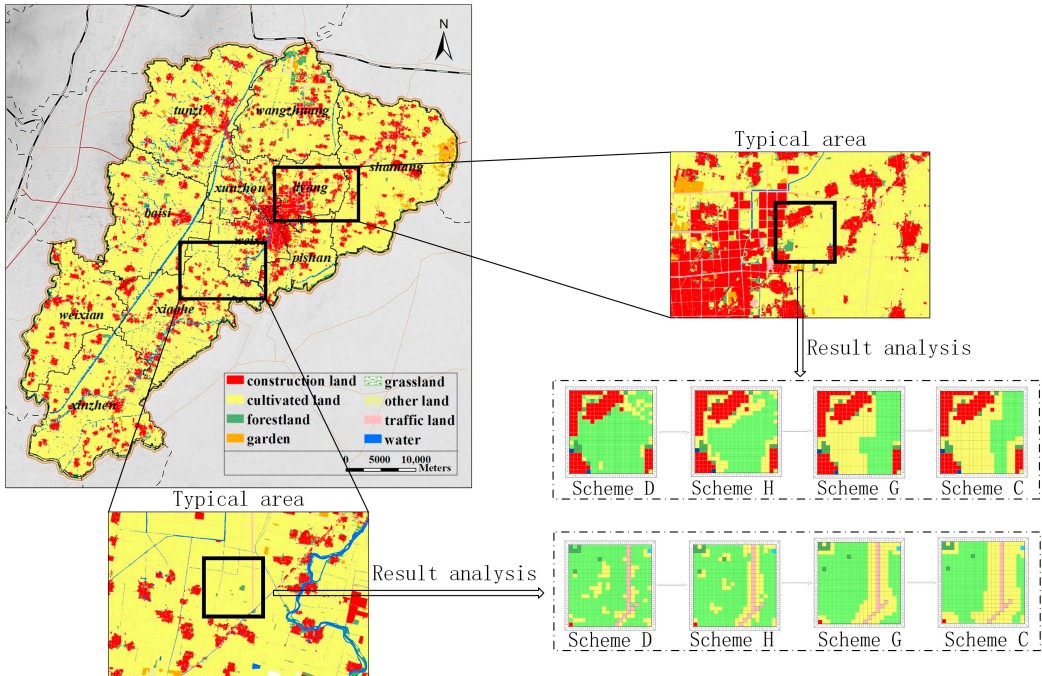

**Figure 7.** Changes in the spatial patterns of typical regions as a result of the stability weights.

In summary, the difference in objective function values produced by the different weight combinations were more obvious. The experiment reflected that the three subobjective functions of the model were mutually restricted. Decision makers can weigh the allocation coefficients according to the need to obtain a suitable result for a specific decision.

*4.3. Analysis of the Impact of Improvement on the Model*

To analyze the impact of the introduction of immune system concepts on the model and to compare the efficiency of particle swarm optimization (PSO), artificial immune algorithm (AIA), and immune particle swarm optimization (AIA-PSO) in solving multi-objective optimization problems, this study conducted experiments from the aspects of model convergence ability, optimization ability, and stabilization ability.

1. Convergence ability of the model

To compare the efficiency of the three models in solving the problem of permanent basic farmland delineation, the experimental results were compared by using multiple test values. Since scheme H was a balanced solution of three subobjective functions, we used this scheme for the experiments.

At the same time, considering that the results had a certain randomness, the three models were run 100 times each, and the average number of iterations and the average convergence times were calculated, as shown in Table 10, and the corresponding convergence curve was plotted as shown in Figure 8.

**Table 10.** Mean convergence times and running times of the three models under scheme H.

| Model | Average Convergence Iterations | Mean Convergence Time (h) |
| --- | --- | --- |
| PSO | 79 | 1.7 |
| AIA | 86 | 2.3 |
| AIA-PSO | 52 | 4.1 |

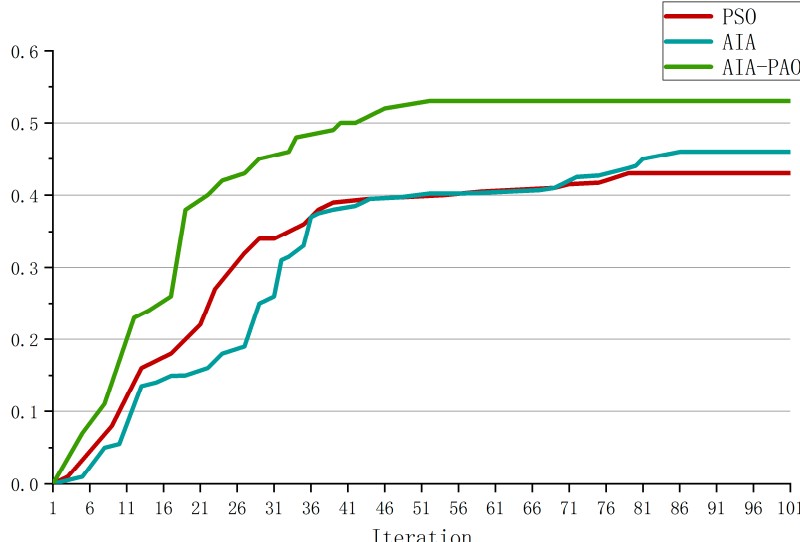

**Figure 8.** Convergence plots of the three models under scheme H.

As seen from the graph and table, the AIA-PSO model required the lowest number of iterations, but the calculation took longer. In terms of the convergence speed, AIA required the most iterations, while AIA-PSO had the lowest number of iterations on average. This showed that PSO had a faster optimization speed compared to the AIA. The introduction of immune system concepts further improved the optimization speed of the AIA-PSO model. From the perspective of running time, the running time required by AIA-PSO was much longer than those of the other two models. This was because it increased the immune process of the AIA during the running process and improved the particle positioning and speed update mechanism, which greatly increased the complexity of the calculation, thereby consuming more time. In general, compared with PSO and AIA, AIA-PSO had a higher convergence efficiency and faster optimization speed, so the loss of time was acceptable.

2. Optimization ability of the model

To compare the optimization capabilities of the three models for the three subobjectives when solving the multi-objective optimization problem, the following experiments were performed in this study: scheme H was still used, and three models were used to solve the three subobjective functions 100 times, and the optimal value results were recorded and averaged, as shown in Table 11.

**Table 11.** Average subobjective function values calculated by the three models.

| Model | $f_s$ | $f_l$ | $f_w$ |
|---|---|---|---|
| PSO | 34,988.8 | 0.9102 | 44,836.75 |
| AIA | 38,268.3 | 0.9273 | 45,397.92 |
| AIA-PSO | 36,163.4 | 0.9561 | 46,934.68 |

It can be seen through comparison of the average values in the table that for the three subobjective functions, the optimization ability of the AIA was better than that of PSO, and the AIA-PSO model had also improved to a certain extent after the introduction of immune system concepts. Especially under the two subobjectives of stability and stability, the optimization ability of AIA-PSO exceeded the PSO and AIA models, and the solving ability was significantly improved. Compared with the PSO model, the average objective function values calculated by AIA-PSO increased by 3.36%, 5.04%, and 4.68%, respectively.

3. Stability of the model

To analyze the stability of the three models, we still chose scheme H, ran the three models 100 times each, calculated the optimal solution of the three subobjective functions under the three models, and plotted the result curve, as shown in Figure 9. The mean and standard deviation were then calculated, as shown in Table 12.

**Table 12.** Means and standard deviations of the calculation results of the three models.

| Model | Mean Value | | | Standard Deviation | | |
|---|---|---|---|---|---|---|
| | $f_s$ | $f_l$ | $f_w$ | $f_s$ | $f_l$ | $f_w$ |
| PSO | 34,984.81 | 0.91006 | 44,836.769 | 174.56 | 0.00406 | 86.948 |
| AIA | 38,255.35 | 0.92736 | 45,703.902 | 277.52 | 0.00682 | 153.377 |
| AIA-PSO | 36,127.13 | 0.95574 | 46,923.043 | 179.63 | 0.00604 | 103.046 |

According to the above calculation results, compared with the average value, the three subobjective function values of the AIA-PSO model were higher than those of the PSO, which indicates that the addition of immune system concepts strengthens the global search ability of the original algorithm, and the optimization ability was better. However, it can be seen from the standard deviations that those of the AIA and AIA-PSO were higher than that of PSO, indicating that the two models were more volatile and that the algorithm was less stable than PSO. From the result curve drawn in Figure 9, we can see more clearly that the stabilization ability of the AIA-PSO model was less effective at solving basic farmland delineation problems than PSO but was more effective than the AIA. The addition of immune operations improved the problem of the PSO falling into a local optimal solution but reduced the stability of the model. Although AIA's optimization ability was higher than that of PSO, the stability of the model is also an important indicator that cannot be ignored when solving multi-objective problems. The AIA-PSO model combined the stability of the PSO and the global search ability of the AIA and can obtain more satisfactory calculation results.

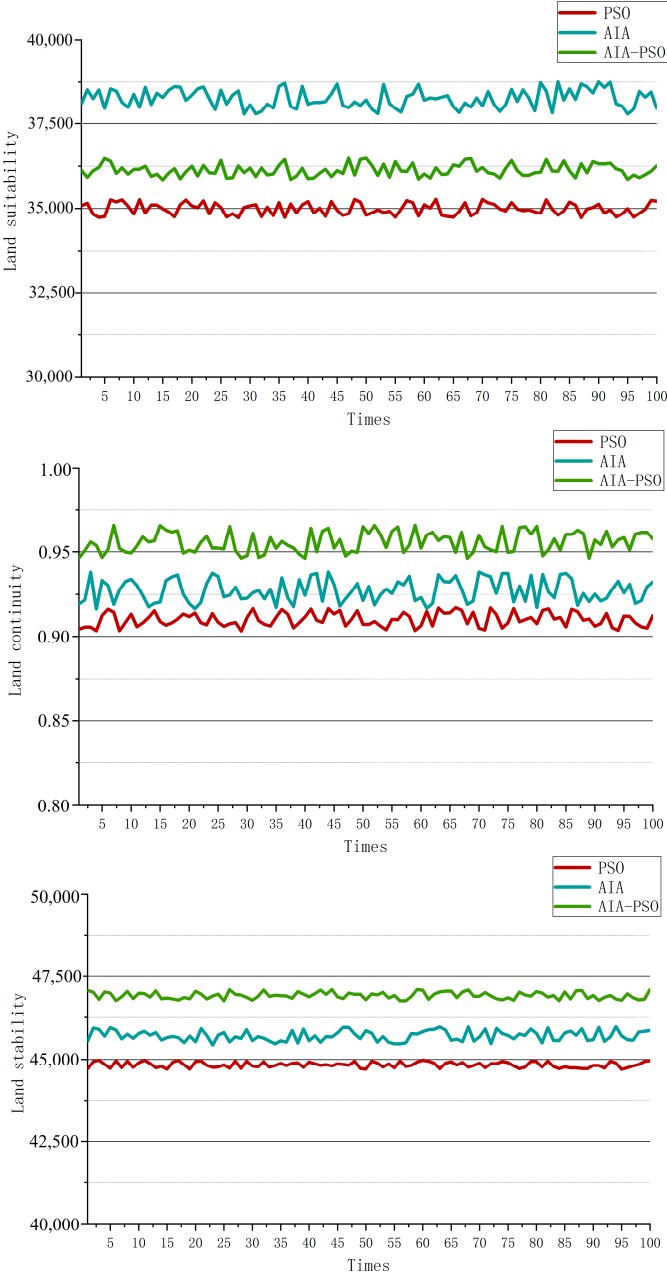

**Figure 9.** Graph of subobjective function values for 100 runs of three models.

### 4.4. Analysis of the Pattern of Permanent Basic Farmland in Xun County

In combination with the comprehensive needs of the model, we chose the abovementioned scheme H to delimit the permanent basic farmland in Xun County. According to the results of the experiment, the area of permanent basic farmland selected by each township and its corresponding grade were counted and recorded. Figure 10 shows the results of the delineation of permanent basic farmland in Xun County. The demarcation results of the permanent basic farmland in Xun County are shown in Table 13.

Due to the randomness of the particle swarm algorithm in searching for a solution space, the distribution of particles becomes more spatially dispersed. After introducing constraint conditions that consider the continuity and stability goals, although the final delineation scheme of the experiment still had some unsatisfactory spatial areas, the overall continuity was already satisfactory. In terms of quality assurance, due to the guidance of the land suitability objective function, the delineated

permanent basic farmland had a specific approach to higher-level farmland. Almost all level five cultivated land and level six cultivated land were classified as permanent basic farmland. In terms of the spatial continuity, the concentrations of land and contiguous land around the neighboring towns and near traffic roads were poor in the study area. The cultivated land in these areas is mostly fragmented and disconnected, so it was not classified as permanent basic farmland. In terms of the stability of the spatial pattern, the land stability objective conflicted with the land use constraints in Section 3.2. The cultivated land around the towns with dense roads played a role in controlling the further expansion of the urbanization to a certain extent, and the remaining areas avoided the township center and the surrounding roads. The experimental results not only ensured the quality of the cultivated land but also ensured the continuous extent and long-term stability of the cultivated land.

It can be seen from the data that Shantang town, Tunzi town, and Xinzhen town had the largest cultivated land area selected for permanent basic farmland. The selected cultivated land area of these three towns was more than 8000 hectares. The three towns all had flat topography, and the cultivated land was contiguous. However, there were problems with insufficient irrigation facilities and traffic congestion. In addition, in some areas of Shantang town and Xinzhen town, the cultivated land had a long cultivation distance, and the quality of cultivated land was low. The irrigation facilities in Weixian town were also lacking, and the protection of cultivated land was poor. Chengguan town is the seat of the Xun County government and is surrounded by Liyang town. It is the county's political center and economic core. The large inflow of population and rapid economic growth have led to the rapid expansion of Chengguan town and complicated roads, so no cultivated land was selected as permanent basic farmland. The results showed that the area designated in Xun County for permanent basic farmland was 60,265.27 hectares, which met the protection target of 60,246.67 hectares. The land had an average grade of 6.32 and a protection rate of 84.89% was achieved, which was 18.6 hectares more than planned.

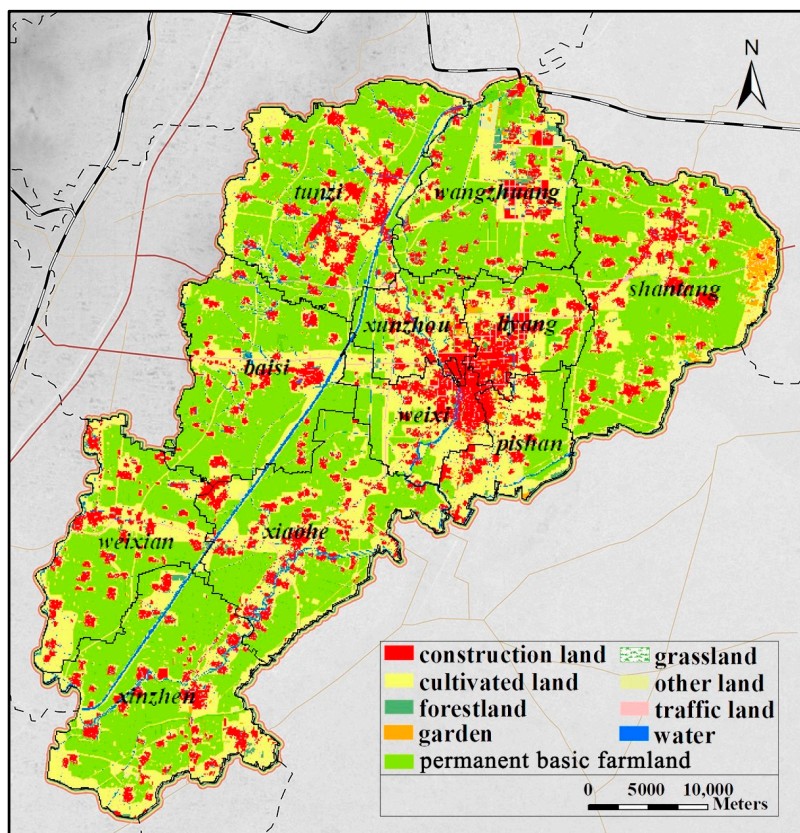

**Figure 10.** Map of the permanent basic farmland protection area in Xun County.

**Table 13.** Statistical table of the delimitation results of the permanent basic farmland in each township of Xun County (ha).

| No. | Administrative Regions | | Type of Cultivated Land | | | Average Grade |
|---|---|---|---|---|---|---|
| | Code | Name of Town | Irrigated | Dry | Subtotal | |
| 1 | 410621100 | Chengguan | 0 | 0 | 0 | |
| 2 | 410621101 | Shantang | 10,827.39 | 0 | 10,827.39 | 7.03 |
| 3 | 410621102 | Tunzi | 9071.26 | 313.53 | 9384.79 | 6.42 |
| 4 | 410621104 | Xinzhen | 8847.94 | 2.69 | 8850.63 | 6.18 |
| 5 | 410621105 | Xiaohe | 7687.47 | 0 | 7687.47 | 6.05 |
| 6 | 410621106 | Liyang | 3190.08 | 0 | 3190.08 | 6.48 |
| 7 | 410621107 | Weixian | 6073.82 | 7.71 | 6081.53 | 5.73 |
| 8 | 410621200 | Wangzhuang | 5858.45 | 4.62 | 5863.07 | 6.15 |
| 9 | 410621201 | Baisi | 8249.82 | 67.27 | 8317.09 | 6.2 |
| 10 | 410621203 | Liyang | 0 | 0 | 0 | |
| 11 | 410621204 | Xun County woodland | 0 | 0 | 0 | |
| 12 | 410621205 | Shantang township woodland | 0 | 0 | 0 | |
| 13 | 410621206 | Xun County farm | 13.81 | 0 | 13.81 | 8 |
| 14 | 410621208 | Original forest farm six team | 11.93 | 0 | 11.93 | 8 |
| 15 | 410621209 | Original forest farm six team | 37.48 | 0 | 37.48 | 6 |
| 16 | 410621210 | Disputed areas of Dongzhu, Zhongzhu and Xizhu | 0 | 0 | 0 | |
| | Total | | 59,869.45 | 395.82 | 60,265.27 | 6.32 |

The permanent basic farmland delimitation model based on the hybrid particle swarm optimization algorithm comprehensively considered the land conditions, spatial patterns, development potential, and other factors, along with constraints such as the area, land use, and terrain, so that the delineation results were able to meet the requirements of optimal farmland quality. The model guaranteed the characteristics of a concentrated and continuous spatial pattern and improved the long-term stability of basic farmland protection areas, effectively guaranteeing regional food security.

## 5. Conclusions

Permanent basic farmland is the basic guarantee of China's food security and an important part of China's strategic security measures. Permanent basic farmland delineation is essentially a multi-objective spatial optimization problem. In this study, based on the three subobjectives of land suitability, land continuity, and land stability, and with the use of particle swarm optimization, the artificial immune algorithm was introduced, and immune operations were added to the algorithm flow. A permanent basic farmland delineation model based on a hybrid particle swarm optimization algorithm was constructed, and the model was verified with Xun County of Henan Province as the research area.

1. The results of the experimental comparative analysis showed that there were conflicts between land suitability, land continuity, and land stability. The improvement in the value of any one subobjective function was at the expense of the other two subobjective values. The corresponding weights $\rho_1$, $\rho_2$, and $\rho_3$ were allocated according to the specific decision objectives to obtain a more satisfactory delineation scheme.

2. With the increase in the weights $\rho_2$ and $\rho_3$ corresponding to the continuity and stability in the experiment, the consistency target guided the particles to find the area where the spatial concentration was formed and gradually produced a more obvious concentration effect. The stability objective caused the selected area to gradually move away from towns and roads from the periphery of the town, reducing the possibility of land occupation due to the renewed expansion of the town.

3. The PSO algorithm has a strong spatial search ability and global optimization ability to solve multi-objective problems. Combined with the spatial processing capabilities of GIS, which reduced

manual intervention and improves work efficiency, it has strong operability and provides a favorable guarantee of permanent basic farmland demarcation results and quality.

4. The global search capability of the artificial immune algorithm was used to compensate for the shortcomings of particle swarm optimization, which easily falls into a local optimal solution. A hybrid particle swarm optimization algorithm was constructed to improve the efficiency of the model. Compared with the original model, the number of iterations was reduced by 34.2%, and the optimization ability of the three subobjective functions was increased by 3.36%, 5.04%, and 4.68%. However, the improved AIA-PSO had 2.4 times the running time of PSO and had large fluctuations.

**Author Contributions:** Conceptualization, H.W. and W.L.; Data curation, W.L.; Formal analysis, W.L. and K.N.; Methodology, H.W., W.L. and W.H.; Experiment and result analysis, H.W. and W.L.; Visualization, W.L. and W.H.; Writing—original draft, H.W. and W.L.; Writing—review and editing, H.W., W.H. and K.N. All authors have read and agreed to the published version of the manuscript.

**Funding:** The Project was Supported by the National Natural Science Foundation of China Grant Number 41601418, the Open Fund of Key Laboratory of Urban Land Resources Monitoring and Simulation, Ministry of Natural Resources, Grant No. (KF-2019-04-038). This research was also funded by Key scientific and technological projects in Henan Province, Grant No. (172102210539).

**Conflicts of Interest:** The authors declare there is no conflicts of interest regarding the publication of this paper.

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
