# Peer review of "A Multi-Objective Permanent Basic Farmland Delineation Model Based on Hybrid Particle Swarm Optimization"

_ijgi, doi:10.3390/ijgi9040243_

Round 1

Reviewer 1 Report

This paper developed a multi-objective permanent basic farmland delineation model using an immune particle swarm optimization algorithm, and considering three sub objectives (land suitability, continuity, and stability) and four constrains ( total area of farmland, current land use, terrain conditions, and town boundaries). The manuscript is thoroughly well written. The references, figures, and tables are relevant and appropriate. The authors have given enough effort to explain model constructions, results and discussion. With the following minor corrections, this manuscript is recommended for publications.

Line 64- 65: why not simply say “scholars’ only as the references tell who the researchers are.

Figure 1 should include three layer of map- china, Henan province and the study county, also, replace the map of Xun County with higher resolution image with better label color and, background color; the labels within map are not clear at all.

In figure 2 (g, h)- why label is 6 decimal place instead of 2 as in Figure.2 (a)

Same applies to figure 6 and 7- higher resolution image should be included as labels are hardly readable.

Author Response

Point 1: Line 64- 65: why not simply say “scholars’ only as the references tell who the researchers are.

Response 1: We have made correction according to the Reviewer’s comments.

Point 2: Figure 1 should include three layer of map- china, Henan province and the study county, also, replace the map of Xun County with higher resolution image with better label color and, background color; the labels within map are not clear at all.

Response 2: Considering the Reviewer’s suggestion, we have modified Figure 1 to include a three-layer map—China, Henan Province, and study county. The label color and background color of the Xun County map have also been modified. We have changed Figure 1 to higher-resolution images.

Point 3: In figure 2 (g, h)- why label is 6 decimal place instead of 2 as in Figure.2 (a)

Response 3: We have unified the number of decimal places in Figure 2.

Point 4: Same applies to figure 6 and 7- higher resolution image should be included as labels are hardly readable.

 Response 4: We have changed Figure 1, Figure 2, Figure 6, and Figure 7 to higher-resolution images.

Reviewer 2 Report

The problem of looking for farmland location is a crucial issue in many countries. Optimization of the farmland location is the basic guarantee for the global food security not only in China- it is a global issue.

Presented a new multiobjective spatial solution revealed that the application of particle swarm optimization (PSO) model give an opportunity to find the best location, taking into consideration many spatial factors.

The content and the structure of the manuscript is appropriate.  Moreover, the language of the article is also appropriate and it do need any more improvements.

Nevertheless, I think that the graphics should be improved. The captions are to small – figures:1-2, 6-7, 10.

After manuscript improvement according to my remarks it can be published.

Author Response

Point 1: I think that the graphics should be improved. The captions are to small – figures:1-2, 6-7, 10.

Response 1: We have changed Figure 1, Figure 2, Figure 6, Figure 7, and Figure 10 to higher-resolution images to make the titles and legends clearer.

Reviewer 3 Report

The topic of the paper and the used approach are interesting and could be successfully implemented in the planning process.

The paper in its actual form needs some improvement before the publication and in particular the structure of the paper itself is not in accordance to the Journal Manuscript preparation criteria https://www.mdpi.com/journal/ijgi/instructions

Then I suggest the authors to rearrange it according to the proper structure: particularly the lack of a Methods section is crucial. Methods used must be clearly described, referring to the proper literature. Actually, some aspects are improperly present in the Introduction section. Some more efforts must be spent by the authors in clearly and briefly describe the method and the applied model.

Please pay attention in properly cite data source through the text.

Authors may find notation in the attached file.

Besides, please reduce Abstract: it should be 200 words long. No detail should be comprised in the abstract, but a general overview of the problem and of the proposed approach.

Please add some more references about Europe in the initial part of the introduction as the dynamic of urbanization and cultivated land have been largely testified. These are some examples (you can consider or not) focusing on Europe, among the many that can be found:

  • Piana, Pietro, et al. "Geomorphological landscape research and flood management in a heavily modified Tyrrhenian catchment." Sustainability 11.17 (2019): 4594.
  • Paliaga, Guido, et al. "Exposure to Geo-Hydrological Hazards of the Metropolitan Area of Genoa, Italy: A Multi-Temporal Analysis of the Bisagno Stream." Sustainability 12.3 (2020): 1114.
  • Acquaotta, Fiorella, et al. "Increased flash flooding in Genoa Metropolitan Area: a combination of climate changes and soil consumption?." Meteorology and Atmospheric Physics 131.4 (2019): 1099-1110.
  • Antrop, M. Landscape change and the urbanization process in Europe. Landsc. Urban Plan. 2004, 67, 9–26.
  • Antrop, M. Changing patterns in the urbanized countryside of Western Europe. Landsc. Ecol. 2000, 15, 257–270.
  • Viciani, D.; Dell’Olmo, L.; Gabellini, A.; Gigante, D.; Lastrucci, L. Landscape dynamics of Mediterranean montane grasslands over 60 years and implications for habitats conservation. A case study in the northern Apennines (Italy). Landsc. Res. 2017, 43, 952–964. 

Finally, English language must be revised, eliminating grammar errors and improving readability and clearness.

Author Response

Point 1: Line 11-37: I think that the graphics should be improved. The captions are to small – figures:1-2, 6-7, 10. Abstract is too long: please reduce it.

Response 1: We have shortened the abstract to less than 200 words. In this part we mainly describe the coping problems and application methods.

Point 2: Line 43: Please add more references: it is a worldwide issue. 

Response 2: Considering the Reviewer ’s suggestion, we have added some more references about other countries in the initial part of the introduction.

Point 3: Line 49-54, 308, 359: Please add proper references. 

Response 3: We have added references where they are missing in the text. 

Point 4: Line 79-95: Please add the reference close to the author’s name.

Response 4: We have moved the reference number closer to the author's name. 

Point 5: Line 110-117: A methods section should be added, to describe the approach, models and analytical methodologies used. 

Response 5: We have modified the article structure based on reviewers' comments, changing Chapter 2 to "Data" and Chapter 3 to "Methodology". The "Methodology" section describes the methods, models and analysis methods used. We have added a paragraph in the introduction to clearly indicate the algorithm used and main work content in this article.

Point 6: Line 128, 154, 402, 429: Please magnify text and particular the legend. 

Response 6: We have changed Figure 1, Figure 2, Figure 6, Figure 7, and Figure 10 to higher-resolution images to make the titles and legends clearer.

Point 7: Line 138-146: Please add proper data references. 

Response 7: We have changed the description of data source for the study area in Section 2.2 to tabular presentation. 

Point 8: Line 173: Please add proper reference. 

Response 8: We have added a new paragraph in this text to mark the source of file "Analysis Report on the Quality of Cultivated Land in Xun County in 2018", "Agricultural Land Quality Grading Regulations" and "Cultivated Land Quality in Henan Province in the Annual Update Evaluation Work Training Materials". 

Point 9: Line 122, 203, 204, 328: Re-write clearly and eliminating the errors. 

Response 9: We have corrected the grammatical or written errors in the text proposed by the reviewers.

Point 10: Line 212, 246: How the weights have been assigned? 

Response 10: we have added specific weight values for α1 ,α2, α3, α4 and σ1,σ2  based on existing research results, and added a reference at corresponding position.